# Cell-Free-DNA-Based Copy Number Index Score in Epithelial Ovarian Cancer—Impact for Diagnosis and Treatment Monitoring

**DOI:** 10.3390/cancers14010168

**Published:** 2021-12-30

**Authors:** Elena Ioana Braicu, Andreas du Bois, Jalid Sehouli, Julia Beck, Sonia Prader, Hagen Kulbe, Bernd Eiben, Philipp Harter, Alexander Traut, Klaus Pietzner, Ralf Glaubitz, Beyhan Ataseven, Radoslav Chekerov, Christoph Keck, Thomas Winkler, Sebastian Heikaus, Peggy Gellendin, Ekkehard Schütz, Florian Heitz

**Affiliations:** 1Department for Gynecology with the Center for Oncologic Surgery Charité Campus Virchow-Klinikum, Charité—Universitätsmedizin Berlin, Corporate Member of Freie Universität Berlin, Humboldt-Universität zu Berlin, and Berlin Institute of Health, 12169 Berlin, Germany; ioana@braicu.de (E.I.B.); Jalid.Sehouli@charite.de (J.S.); hagen.kulbe@charite.de (H.K.); klaus.pietzner@charite.de (K.P.); radoslav.chekerov@charite.de (R.C.); peggy.gellendin@charite.de (P.G.); 2Department for Gynecology and Gynecologic Oncology, Kliniken Essen-Mitte, 45136 Essen, Germany; Prof.duBois@googlemail.com (A.d.B.); Sonia.Prader@sabes.it (S.P.); p.harter@gmx.de (P.H.); a.traut@kem-med.com (A.T.); ataseven@gmx.net (B.A.); 3Chronix Biomedical GmbH, 37079 Goettingen, Germany; beck@liquidbiopsy.center (J.B.); esc@chronixbiomedical.de (E.S.); 4Liquid Biopsy Center GmbH, 37075 Goettingen, Germany; eiben@eurogen.de (B.E.); glaubitz@liquidbiopsy.center (R.G.); thomas.winkler@amedes-group.com (T.W.); 5Department of Obstetrics and Gynecology, General Hospital (SABES-ASDAA), 39042 Brixen-Bressanone, Italy; 6Department of Obstetrics and Gynecology, Innsbruck Medical University, 6020 Innsbruck, Austria; 7Amedes Genetics, 30625 Hannover, Germany; christoph.keck@amedes-group.com; 8Department of Obstetrics and Gynecology, University Hospital, LMU Munich, 81675 Munich, Germany; 9Center for Pathology, Kliniken Essen-Mitte, 45122 Essen, Germany; s.heikaus@pathologie-essen.de

**Keywords:** epithelial ovarian cancer, high-grade, debulking surgery, liquid biopsy, disease monitoring, tumor burden

## Abstract

**Simple Summary:**

The prognosis of ovarian cancer is dependent on the tumor stage and the development of chemotherapy resistance. Using low-coverage cell-free tumor DNA sequencing, we were able to determine the chromosomal instability (CI) of tumors that are frequently found in patients with primary advanced and recurrent high-grade ovarian cancer from a blood sample. We were able to show that the CI could be used for the reliable detection of ovarian cancer in comparison to healthy controls. Moreover, we showed that the CI was significantly associated with the prognostic and predictive clinical measures in primary and recurrent ovarian cancer. The high diagnostic accuracy of the tumor CI derived from cfDNA analysis might lead to the optimization of main prognostic determinants in patients with ovarian cancer. As the CI is a characteristic feature in high-grade ovarian cancer, no upfront tumor tissue analysis is required to identify genomic alterations for targeted sequencing of cfDNA, if the herein described low-coverage sequencing and CNI-Score determination is used.

**Abstract:**

Background: Chromosomal instability, a hallmark of cancer, results in changes in the copy number state. These deviant copy number states can be detected in the cell-free DNA (cfDNA) and provide a quantitative measure of the ctDNA levels by converting cfDNA next-generation sequencing results into a genome-wide copy number instability score (CNI-Score). Our aim was to determine the role of the CNI-Score in detecting epithelial ovarian cancer (EOC) and its role as a marker to monitor the response to treatment. Methods: Blood samples were prospectively collected from 109 patients with high-grade EOC. cfDNA was extracted and analyzed using a clinical-grade assay designed to calculate a genome-wide CNI-Score from low-coverage sequencing data. Stored data from 241 apparently healthy controls were used as a reference set. Results: Comparison of the CNI-Scores of primary EOC patients versus controls yielded sensitivities of 91% at a specificity of 95% to detect OC, respectively. Significantly elevated CNI-Scores were detected in primary (median: 87, IQR: 351) and recurrent (median: 346, IQR: 1891) blood samples. Substantially reduced CNI-Scores were detected after primary debulking surgery. Using a cut-off of 24, a diagnostic sensitivity of 87% for primary and recurrent EOC was determined at a specificity of 95%. CNI-Scores above this threshold were detected in 21/23 primary tumor (91%), 36/42 of platinum-eligible recurrent (85.7%), and 19/22 of non-platinum-eligible recurrent (86.3%) samples, respectively. Conclusion: ctDNA-quantification based on genomic instability determined by the CNI-Score was a biomarker with high diagnostic accuracy in high-grade EOC. The applied assay might be a promising tool for diagnostics and therapy monitoring, as it requires no a priori information about the tumor.

## 1. Introduction

Primary epithelial ovarian cancer is the 11th most common cancer, the 5th leading cause of death of cancer in women, and is the leading cause of death in gynecologic cancers in Western countries [1]. High mortality rates are caused by late diagnosis with advanced tumor stages in up to 75% of patients and frequent development of resistance to platinum-based chemotherapy. Standard treatment consists of debulking surgery that aims at macroscopic complete resection, followed by chemotherapy (CTX) with carboplatin/paclitaxel [2]. Primary EOC is highly chemotherapy sensitive; nevertheless, at least 50% of patients will recur and will subsequently die due to metastatic disease. More recently, several maintenance therapies have been introduced that lead to a delayed occurrence of relapses or even prolonged long-term survival. Maintenance treatment with either bevacizumab [3], or bevacizumab and olaparib [4], or olaparib [5], or niraparib [6] is indicated based on a multifactorial selection process, including *BRCA* mutational status, homologous recombination deficient (HRD) status, stage and size of residual disease, histological type, patients’ characteristics and co-morbidities, and local approval status. 

In recent years, cell-free DNA (cfDNA) has attracted increasing attention and application as a non-invasive biomarker for several cancer entities. The spectrum of diagnostic applications includes cancer screening, the detection of specific tumor mutations relevant for targeted therapies, and the quantification of tumor-derived cfDNA (ctDNA) to monitor treatment success. The detection of copy number differences is particularly useful for monitoring treatment since most tumors exhibit a high degree of chromosomal instability and, thus, the applicability may be broader than monitoring driver mutations, which usually only occur in certain tumor types. Hence, the determination of copy number differences across the entire genome is more suitable as a universal marker. Furthermore, copy number differences can be detected using low-coverage whole-genome sequencing, which is cost-effective and is less prone to false-positive findings and quantification errors due to sequencing errors. Moreover, it requires less extensive bioinformatic processing of the raw data [7,8]. The goal of the current study was to verify and extend data on the application of the copy number index (CNI) score described earlier by some of our group [9,10,11,12] for patients with primary and recurrent high-grade EOC.

## 2. Materials and Methods

This prospective multicenter cohort study was conducted in the Department for Gynecology and Gynecologic Oncology of Kliniken Essen-Mitte (KEM) and Department for Gynecology with the Center for Oncologic Surgery, Campus Virchow-Klinikum, Charité, Germany, Berlin (Charitè). The study was approved by the external review board of the Ärztekammer Nordrhein (lfd. no. 2018342) for KEM at the 21 December 2018 and by the Charitè Ethical Committee (EA1/235/18) at the 31 January 2019. 

The goal of the present study was to prospectively assess the performance of a tumor-uniformed liquid biopsy assay in patients with high-grade EOC. A power analysis according to the primary goal was performed with the following assumptions: mean difference from normal cohort: twofold with a sigma of 4 and an alpha of 0.01, which resulted in a minimum of 66 patients to reach a power of 0.95. The following inclusion criteria had to be fulfilled: patients with primary advanced (FIGO IIIB–IVB), platinum-eligible or platinum non-eligible relapsed high-grade EOC, and a signed informed consent form. 

Patients were not included in the study if their tumors were non-high-grade, non-invasive, or non-epithelial. Patients in the primary advanced high-grade group could be included either before primary debulking surgery, defined as the “upfront group”, or after the onset of first-line chemotherapy until the end of first-line chemotherapy, defined as the “CTX group”. In addition, the “follow-up group” consisted of patients after the end of first-line chemotherapy that had no signs of recurrent disease. High-grade histologies in the primary advanced high-grade group were determined by gynecologic pathologists using standard HE and immunohistochemistry (including p53). Pathologies from the recurrent cases were derived using chart reviews. *BRCA* statuses from patients were extracted from chart reviews and could include germline or somatic testing. Definitions of platinum eligibility were derived from the ESMO-ESGO consensus conference recommendations on ovarian cancer [13]. Patients’ and tumor characteristics were correlated with a single determination of the copy number index (CNI) score assay from each patient’s plasma. 

In patients who finished systemic therapy and who were in the follow-up group, no regular CT scan or CA125 analyses were recommended. In addition, stored data from 241 plasma samples from apparently healthy volunteers (120 female and 121 male) that were partially published in previous studies [9,10] were used as a comparison set. The median age of this control group was 50 years (<50 years: 50%; 50–59 years: 22%; 60–69 years: 20%; >70 years: 8%). The CNI-Score values in the group of healthy controls did not show an association with age or gender (*p* > 0.05), hence the values from the entire control group were used for comparison.

### 2.1. Liquid Biopsy Assay

The cell-free tumor DNA (ctDNA) assay is an accredited (ISO/EN 14189:2018) clinical-grade assay that is based on the quantification of ctDNA by using copy number variances (a hallmark of malignant cells) detected in the plasma of patients using shallow sequencing [9]. The CNI-Score test is performed while tumor uninformed and is proportional to the amount of circulating tumor DNA in the plasma of a patient.

### 2.2. cfDNA Extraction and Sequencing

Blood samples (2 × 8–10 mL) were collected in white blood cell stabilizing vacutainer tubes (Cell-free DNA BCT, Streck Inc., Omaha, NE, USA) and were shipped to the Liquid Biopsy Center GmbH laboratories. Plasma was separated and remaining cell debris was removed using dual centrifugation. cfDNA was extracted using the High Pure Viral Nucleic Acid Large Volume Kit (Roche Diagnostics, Mannheim, Germany) and the total cfDNA yield was determined using a digital droplet PCR assay, as described before [9]. Sequencing libraries were prepared from at least 10 ng cfDNA using the ThruPlex DNA Seq Kit (Takara, Shiga, Japan) according to the manufacturer’s instructions. Sequencing was conducted on a NextSeq500 (Illumina, San Diego, CA, USA) system. Each sequencing batch consisted of 17 patient samples in addition to one positive and one negative control sample; an average of 27.5 M (SD: 8.75 M) read pairs were generated per sample.

### 2.3. CNI-Score Calculation

CNI-Scores were calculated as described previously [9]. Briefly, the obtained sequencing reads were mapped to the human reference genome (HG19). An average of 27 M (SD: 8.6 M) mapped read pairs were analyzed per sample. Read counts of fragments with an insert size of <170 bp were obtained for sliding autosomal genomic windows with an average size of 0.55 Mbp. After correction for GC and mappability biases, the read counts were transformed to log_2_ ratios using the R package (QDNAseq). Signals were condensed by combining 10 windows to yield an average final window size of 5.5 Mbp and 701 final windows (average reads ~18,000 per bin). For each window, a Z-score was calculated against the mean and standard deviation obtained from 137 normal samples. Windows with Z-scores of >2.84 or <−2.84 (critical difference, defined as −2 × sqrt(2)) were considered as significantly different from the diploid copy number state and absolute scores of such windows were summed to give the final CNI-Score.

Total cfDNA was quantified as genomic copies per milliliter plasma (cp/mL) using a digital droplet PCR assay, as described elsewhere [14].

### 2.4. Statistical Analyses

Samples were divided according to the patients’ clinical disease status in the following groups: (i) CNTRL: control group without malignant disease; (ii) “upfront group”: samples obtained prior to primary debulking surgery (exclusively PDS, no interval debulking allowed) at the time of primary diagnosis; (iii) “CTX group”: samples obtained after surgery but not later than 42 days after the end of the 1st-line CTX (min: 2 days, max: 147 days after surgery); (iv) “follow-up group”: samples obtained later than 42 days after the end of the 1st-line CTX (min: 169 days, max: 1976 days after surgery); (v) “platinum eligible”: samples collected after a clinically diagnosed recurrence of platinum-eligible tumors prior to the first cycle of a new therapy regimen; (vi) “platinum non-eligible”: samples collected after a diagnosed recurrence of platinum-refractory tumors prior to the first cycle of a new therapy regimen. ROC curve analyses were conducted using the upfront group against the apparently healthy control group using the R package “pROC”.

Non-parametric Wilcoxon rank-sum tests were performed for group comparisons using the rstatix R-package. The *p*-values were adjusted for multiple testing according to the Benjamini–Hochberg method [15].

### 2.5. Comparison of the Copy Number Profiles Found in cfDNA with Tumor Tissue Data (TCGA)

Frequencies of the cfDNA samples that had an amplification or deletion in a given genomic window were plotted using the R package copy number version 1.28.0 using Z-scores of 2.84/−2.84 as thresholds. The post-operative samples were excluded from this analysis. For comparison of the frequencies in ovarian tumor tissue copy number, data of 585 ovarian serous cystadenocarcinoma samples were downloaded from the TCGA database (https://www.cbioportal.org/study/cnSegments?id=ov_tcga_pan_can_atlas_2018, 20 August 2020) and plotted with log_2_ ratio thresholds of 0.3/−0.3.

### 2.6. Generation of Copy Number Profile in Selected Tumor Samples

Tumor tissue was analyzed from 5 patients. Cellular DNA was extracted either from FFPE or frozen tissue using the GeneRead DNA FFPE (Qiagen, Hilden, Germany) and DNAeasy Blood and Tissue Kit (Qiagen), respectively. Library preparation and sequencing were conducted as described above. Copy number profiles were calculated using the QDNASeq R package [16].

## 3. Results

Between March 2018 and November 2020, a total of 109 patients were included. Of these, 23 patients were in the upfront group with treatment-naïve advanced disease, 9 patients were in the CTX group, and 13 patients were in the follow-up group. Additionally, 42 patients with platinum-eligible recurrent disease and 22 patients with platinum non-eligible recurrent EOC were included. The median age of all patients was 62 years, ranging between 22 and 84 years; further patient characteristics are presented in Table 1. 

To determine the possible impact of the CNI-Score for ovarian cancer diagnostic purposes, the CNI-Scores of a healthy control group were compared with the 23 patients from the upfront group with high-grade EOC. The median CNI-Score in the healthy control group (*n* = 241) was 9 (interquartile range (IQR): 9), which is in line with the expected statistical value for 701 genomic windows (bins), each with an a priori false-positive probability of 0.05%, as defined by the complementary cumulative probability for the Z-score cut-offs of 2.84/−2.84. Compared with the healthy controls, significantly elevated CNI-scores were found in the upfront group (median: 138, IQR: 754, *n* = 23) (Figure 1A). 

ROC curve analysis of the healthy controls versus the upfront group yielded an optimal cut-off (Youden index) of 24.5 with a corresponding specificity/sensitivity of 95% and 91%, respectively. The 99.6% percentile of the control group, which was suggested as the ideal fixed specificity for ovarian cancer screening by others before [17] corresponded to a CNI-Score of 33 (Table 2).

To determine the possible impact of the CNI-Score for ovarian cancer follow-up purposes, the CNI-Scores of the upfront group (median: 138, IQR: 754, *n* = 23) were compared with the CNI-Scores of the patients in the CTX group (median: 95, IQR: 220, *n* = 9) and the follow-up group (median: 430, IQR: 1.005, *n* = 13). A trend was found for higher CNI-Scores in the follow-up group compared with the CTX group, although the difference between the groups was not significant when adjusted for multiple testing (*p* = 0.06 adj./*p* = 0.041 unadj.) (Figure 1B). The follow-up group included three patients with CNI-Scores below the cut-off and one patient that was collected 1814 days (5 years) after their CTX. It might be assumed that the patient was completely cured, which was also reflected by their CNI-Score of 3. Tumor progression was clinically diagnosed in three patients of the follow-up group with CNI-Scores above the 95th percentile of the normal group (cut-off of 24). In four additional patients with elevated CNI-Scores, progression was not clinically diagnosed until the end of the study and after follow-up intervals between 392 and 601 days. 

To determine the possible impact of the CNI-Score for ovarian cancer treatment-management purposes, the CNI-Scores of patients from the upfront group were analyzed and compared with patients with platinum-eligible and platinum-non-eligible recurrent disease. No significant differences in the CNI-Scores were detected for *BRCA*wt (median: 87, IQR: 225, *n* = 5) versus *BRCA* deficient primary tumors (median: 250, IQR: 1214, *n* = 12) (Figure 2A).

Similarly, no significant differences were found between the CNI-Scores of FIGO stages III (median: 134, IQR: 314, *n* = 8) and IV (median: 207, IQR: 1386, *n* = 12) (Figure 2B) or patients undergoing complete resection (median: 209, IQR: 750, *n* = 11) vs. patients with macroscopic residual disease after surgery (median: 106, IQR: 118, *n* = 8) (Figure 2C).

The median CNI-Scores numerically increased from the upfront group (median: 138, IQR: 754, *n* = 23) to patients with platinum-eligible recurrent tumors (median: 318, IQR: 1.761, *n* = 42) to patients with platinum-non-eligible recurrent tumors (median: 586, IQR: 2.704, *n* = 22), albeit the differences between the groups were not statistically significant (Figure 1C). With the cut-off of 24, a diagnostic sensitivity of 87% for primary and recurrent EOC was determined. CNI-Scores above this threshold were detected in 18/23 of the upfront group cases (78%), 35/42 of the platinum-eligible recurrent samples (83.3%), and 19/22 of the non-platinum-eligible recurrent samples (82.6%). One possible reason for a false-negative CNI-Score result is the absence of copy number aberrations in the tumor genome. Therefore, we investigated the tumor tissue obtained from five patients from the upfront group who had pre-surgery CNI-Scores < 33. After the low coverage shotgun sequencing bins with significantly deviant read-counts were identified by calculating the Z-scores against data obtained from white blood cell nuclear DNA. We could detect copy number differences in 4/5 tumor tissues, indicating that the negative CNI-Scores in these four samples were instead caused by low ctDNA concentrations in the plasma, while one negative CNI-Score was instead attributable to the absence of copy number alterations in the tumor tissue. We further aimed to analyze the similarity of the CNI-Score profiles with the well-described copy number profiles of ovarian tumor tissues available in the TCGA database. The copy number profiles detected in the plasma of patients bearing primary or recurrent tumors were strikingly similar to those detected in ovarian tumor tissue (Figure 3).

The CNI-Score results were also compared with other cfDNA characteristics. No significant correlation between the measured CNI-Scores and the total concentration of cell-free DNA in the plasma of tumor patients was found (Pearson’s R = 0.17, *p* = 0.103). However, a clear relationship between the CNI-Score—i.e., the amount of circulating tumor DNA—and the fragment length of the cfDNA could be demonstrated (Figure 4).

## 4. Discussion

Data presented within this manuscript add evidence to the literature on the diagnostic role of ctDNA-based analyses in patients with high-grade EOC [18,19]. On the one hand, it was shown that this low coverage DNA sequencing approach regarding cfDNA was associated with high sensitivity and specificity to distinguish between EOC and healthy control cases. On the other hand, the presented data give rise to the potential application of the CNI-Score in the management of patients with high-grade EOC. CNI-Score levels were elevated in high-grade ovarian cancers, independent of the time of diagnosis (primary vs. relapse), sensitivity to platinum-based chemotherapy, *BRCA* status, or the outcome of primary debulking surgery (no gross residual mass vs. macroscopical residual mass), underlying its ubiquitarian presence in ovarian cancer patients. The well-described association of ctDNA levels with tumor burden across a variety of different cancer types leads to increasing recognition of ctDNA as a tumor marker [17]. The increasing literature on the performance of ctDNA-based tumor detection shows a widely varying tumor detection rate of 30–100% [20,21,22,23]. The detectability of ctDNA and, thus, the diagnostic sensitivity of the marker depends not only on the quality and technical soundness of the detection method but also on biological factors, such as (metabolic) tumor volume and stage, the cell growth/turnover rate, and the shedding rate of the apoptotic tumor DNA into the bloodstream that can vary between individual tumors [24]. On the technical side, challenges are posed by the low concentration and high fragmentation of ctDNA and the often-found high heterogeneity of tumor cell genomes [25]. The clinical-grade assay applied in the analyses described here uses a low coverage sequencing approach to detect somatic genomic instability that is considered a hallmark of cancer. This is, therefore, particularly suitable as a tumor-specific analytical target [26], whereas no information about the tumor’s genetic makeup is needed. This means the applied assay does not require a priori information about specific tumor mutations and evaluates the whole genome instead of only prespecified target sites [11]. With the data presented, we confirmed the reported high sensitivities to distinguish between high-grade EOC and healthy controls. With the integration of just 23 patients with a primary diagnosis of high-grade EOC, we were able to differentiate between those patients and healthy controls with a sensitivity of 91% at a specificity of 95%. When adding patients with a recurrent disease to the patients with a primary diagnosis of high-grade EOC, the sensitivity and specificity were 87% and 95%, respectively. Other cfDNA-based assays that also target copy number differences reported by other groups yielded sensitivities of 40.6% and 78% at specificities of 93.8% and 99.6%, respectively [18,19]. Furthermore, SNP-targeting high-throughput sequencing approaches are deemed ultrasensitive and were able to detect tumor-specific SNPs in the circulation of 78% of ovarian cancer patients with tumor stages III and IV but are often more time-consuming, costly, and computationally intensive [22]. In addition, ctDNA seems to have better diagnostic accuracy and especially higher specificity than CA-125 and/or RMI (risk of malignancy) scoring, albeit not proven in the present paper, and others propose it as a presurgical biomarker to differentiate benign adnexal masses from a malignant disease [18]. Moreover, follow-up and treatment might be altered in patients with high-grade EOC due to CNI-Score analyses. Our study showed that the CNI-Scores decreased with the reduction of tumor mass by primary surgery and increased again with longer intervals between surgery and the first-line chemotherapy, as was the case with the probability of tumor recurrence. In addition, recurrent diseases were detected in three patients with a CT scan and with an elevated CNI before CA125 was elevated. Provided that these data are validated in a larger study, this observation might lead to another follow-up strategy for patients after finishing their first-line chemotherapy. The “Rustin trial” changed the follow-up standard after the end of the first-line chemotherapy by eliminating the regular measurements of CA125, as early detection of relapse followed by early onset of systemic treatment did not improve survival [27]. However, most recently, the AGO-DESKTOP III showed that surgery for the first recurrence of EOC leads to improved overall survival [28]. It has not been investigated so far whether earlier detection of recurrence might lead to higher rates of complete macroscopic resection at secondary cytoreductive surgery. Nevertheless, it might be of value to set up a trial investigating the role of earlier detection of recurrent disease with the CNI compared to standard measures. 

ctDNA determination might also be included in treatment monitoring and for treatment decisions. A retrospective study of 51 patients with recurrent EOC showed that pre-treatment TP53 mutational levels in ctDNA and a decrease of the TP53 MAF > 60% between baseline and the second cycle of chemotherapy was associated with increased time to progression [29]. Interestingly, the reported sensitivity was 82% in relapsed patients, which is close to what we report herein. In a recent publication using methylation markers to detect ctDNA, a smaller group of ovarian cancers was reported, also showing very similar sensitivities [30]. CNI-Score analysis might also be used for these kinds of measurements in patients undergoing systemic treatment. Previous studies showed that an early decline in the CNI-Score predicted a clinically assessed response to checkpoint inhibitor treatment of patients with advanced cancers [10]. Due to the high costs and severe toxicities of some drugs, early knowledge of effectiveness would help to tailor optimal treatments for patients and reduce the financial burden. In the present study, 84/89 patients with a detectable tumor burden showed elevated CNI-Scores. Therefore, CNI-Score determination at the onset of a new systemic treatment could be a valid method to early evaluate response during treatment. Patients with early response determined by CNI-Score testing would continue treatment, whereas, in patients with no response based on the CNI-Score, treatment would be ceased and new treatment options could be considered. This approach would produce timely, cost-effective results. Both are applicable with the CNI-Score, as the turnaround time between the blood draw and result reporting is about five days, with a total of three days for the wet lab analysis. The cost for the CNI-Score determination is about 10–20% of the cost of one cycle of therapy with, e.g., a checkpoint inhibitor.

The limitations of our study were the lack of available blood samples from patients with early-stage primary tumors, longitudinal blood samples of each patient, and the small sample size of patients after primary surgery and prior to clinical diagnosis of recurrence. However, to the best of our knowledge, this is the largest study so far demonstrating the previously reported high diagnostic sensitivity of chromosomal instability in cfDNA. Since we show herein that the CNI-Score is generally applicable to EOC, the data give rise to future studies that will aim to show the versatility of the CNI-Score in other entities (e.g., non-high grade EOC) and in comparison to established markers (e.g., to other tumor markers, such as CA125) for therapy monitoring and the diagnosis of relapse in EOC. New markers for diagnostics and therapy monitoring that can show incremental value to current standard options might be useful tools in patient management if they outperform current standard options.

## 5. Conclusions

ctDNA quantification based on genomic instability determined by the CNI-Score is a biomarker with high diagnostic accuracy in high-grade EOC. The applied assay might be a promising tool for diagnostics and therapy monitoring, as it requires no a priori information about the tumor.

## Figures and Tables

**Figure 1 cancers-14-00168-f001:**
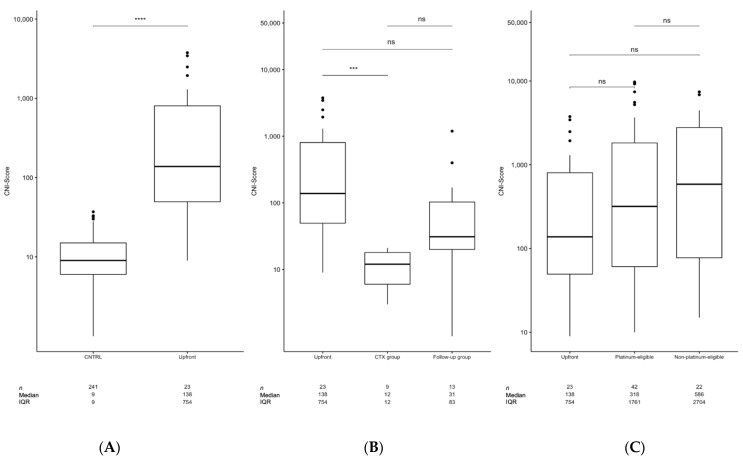
(**A**) Comparison of the CNI-Scores obtained from apparently healthy controls and the respective patient groups. (**B**) Comparison between the upfront group and the CTX group. (**C**) CNI-Scores from the upfront, the platinum-eligible, and platinum-non-eligible recurrence groups of patients. The centerlines show the medians; box limits indicate the 25th and 75th percentiles, as determined using a non-parametric Wilcoxon test (R); whiskers extend 1.5 times the interquartile range from the 25th and 75th percentiles; and outliers are represented by dots. The number of sample points per group (n), median, and interquartile range (IQR) are given as tables. A non-parametric Wilcoxon test was performed to test for statistical differences between the groups (ns: *p* > 0.05, ***: *p* ≤ 0.001, ****: *p* ≤ 0.0001) and adjusted for multiple testing.

**Figure 2 cancers-14-00168-f002:**
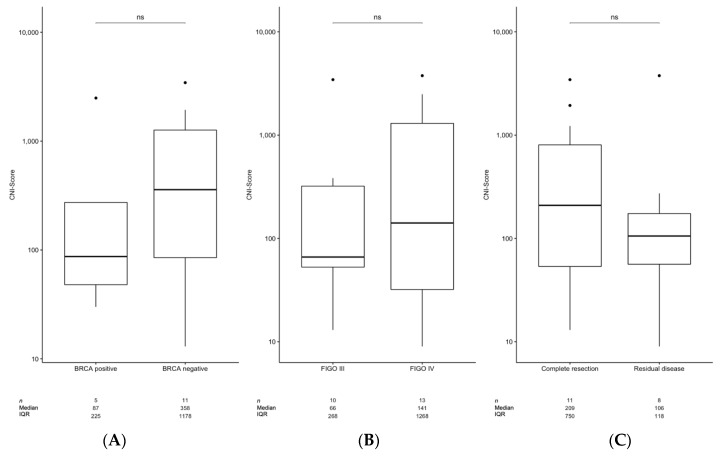
Comparison of the CNI-Scores between (**A**) different FIGO states and (**B**) *BRCA*-mutation-status-only samples from patients from the upfront group were included. (**C**) CNI-Score differences between patients with and without complete resection. The centerlines show the medians; box limits indicate the 25th and 75th percentiles, as determined using a non-parametric Wilcoxon test (R); whiskers extend 1.5 times the interquartile range from the 25th and 75th percentiles; and outliers are represented by dots. The number of sample points per group, median, and interquartile range (IQR) are given as tables. A non-parametric Wilcoxon test was performed to test for statistical differences between the groups (ns: *p* > 0.05).

**Figure 3 cancers-14-00168-f003:**
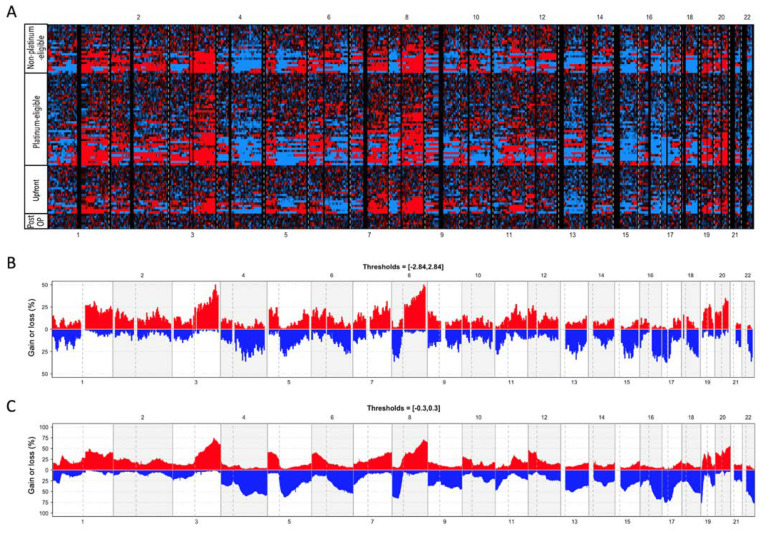
(**A**): Heatmap image displaying the genomic bins with significantly altered copy numbers in the different groups of cancer patients (cut-offs: Z-scores > 2.84 (red), Z-scores < −2.84 (blue)). (**B**): Frequencies of copy number changes detected in the plasma of patients bearing primary or recurrent tumors. (**C**): Frequencies of gains and losses in tumor tissue data obtained from the TCGA database.

**Figure 4 cancers-14-00168-f004:**
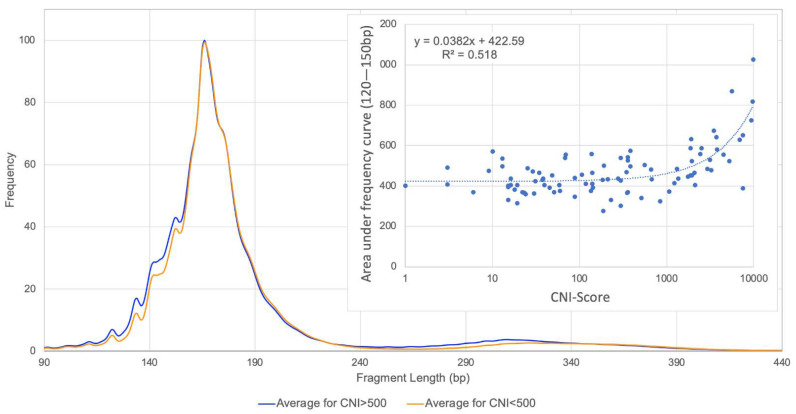
Length distribution of the total cfDNA fragments showed significant differences in samples with CNI-Scores of >500 versus <500 (*t*-test, *p* = 0.00017). The area under the frequency curve between 120 bp and 150 bp was significantly correlated with the CNI-Score levels and, therefore, with the amount of circulating tumor DNA.

**Table 1 cancers-14-00168-t001:** Patients’ characteristics.

Parameter	All *N* = 109 (%)	Upfront Group *N* = 23 (%)	CTX Group *N* = 9 (%)	Follow-Up Group *N* = 13 (%)	Platinum- Eligible Recurrence *N* = 42 (%)	Platinum-Non-Eligible Recurrence *N* = 22 (%)
Median age (years)						
<50	9 (8.2)	4 (17.3)	1 (18)	1 (8.3)	1 (2.4)	2 (9.1)
50–59	34 (31.2)	5 (21.7)	3 (33.3)	1 (8.3)	16 (38.1)	8 (36.4)
60–69	42 (38.6)	12 (52.2)	4 (44.4)	6 (18)	13 (31)	8 (36.4)
≥70	24 (22.0)	2 (8.7)	1 (11.1)	5 (38.5)	12 (28.6)	4 (18.2)
FIGO						
I	n.a.	0	0	1 (8.3)	n.a.	n.a.
II		0	0	0		
III		10 (43.5)	6 (66.6)	8 (66.7)		
IV		13 (56.5)	3 (33.3)	4 (30.8)		
ECOG-PS						
0	88 (80.7)	22 (95.6)	6 (66.6)	9 (75)	33 (78.6)	18 (81.8)
1	8 (7.3)	1 (4.3)	1 (11.1)	0	4 (9.5)	2 (9.1)
>1	2 (1.8)	0	2 (22.2)	0	0	0
x	11 (10.1)	0	0	4 (30.8)	5 (11.6)	2 (9.1)
*BRCA* Mutation						
Yes	33 (30.3)	12 (52.2)	5 (55.5)	6 (18)	11 (26.2)	10 (45.4)
No	53 (48.6)	5 (21.7)	2 (22.2)	4 (33.3)	22 (52.4)	11 (18)
Unknown	23 (21.1)	6 (26.1)	2 (22.2)	3 (23.1)	9 (21.4)	1 (4.5)
Histology						
HG serous	106 (96.3)	23 (100)	9 (100)	13 (100)	41 (95.2)	20 (90.9)
HG endometrioid	2 (1.8)	0	0	0	0	2 (9.1)
Clear-cell	1 (0.9)	0	0	0	1 (2.4)	0

FIGO: Fédération Internationale de Gynécologie et d’ Obstétrique; HG: high-grade; ECOG-PS: Eastern Cooperative Oncology Group-performance score.

**Table 2 cancers-14-00168-t002:** Sensitivity and specificity for detecting ovarian tumors at various CNI-Score cut-offs.

CNI-Score	Specificity (%)	Sensitivity (Primary and Recurrent) (%)	Sensitivity (Primary Only) (%)
24	95	87	91
27	97.5	86	87
31	99	84	83
33	99.6	83	78
37	100	80	78

## Data Availability

The data presented in this study are available on request from the corresponding author. The data are not publicly available due to the preliminary character of the study design without conclusive conclusions.

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
