# Peer review of "Cell-Free-DNA-Based Copy Number Index Score in Epithelial Ovarian Cancer—Impact for Diagnosis and Treatment Monitoring"

_cancers, 2021, doi:10.3390/cancers14010168_

Round 1

Reviewer 1 Report

I am impressed by the work in this manuscript and am recommending it for publication after revising modifications.  I have indicated where certain textual are needed as follows. The authors need to provide the power analysis for this study.  Please explain why the control results in the left and middle panels of Figure 1 seem different. I recommend that they enlist the assistance of a skilled English-speaking editorial writer. I would like to see a little more discussion of how this approach could be used in monitoring treatment, including costs in time and money. 

Detail revision:

  1. line 33: pact of the CNI-Score to detect epithelial ovarian cancer (EOC) and monitor treatment, respectively.

please change "and" into "so that it could be used to"

2.Line 79-81: Please rewrite

3.line 116-119: Please re-write this sentence so that it makes sense.

4.Line 91: primary advanced  Please rewrite

5.Line 92: num-eligible, or non-platinum eligible relapsed high-grade EOC  Please rewrite

6.Line 99: Please check word “were”

7.Line 101: Please check word “were”

8.Line 110: Please check word “were”

9.Line 111: Please check word “does”

10.Line 116-118: Please re-write this sentence so that it makes sense.

11.Page 5: Please enter percentage numbers

12.Line 240: Please check: one patient that was collected 1814 days (5 years) after CTX

13.Line 244: Please check: progression was not clinically diagnosed

14.Line 279: Please delete "rather"

15.Line 304-305: Please re-write this sentence and indicate how.: On the other hand, presented data give rise for potential application of the 304 CNI-Score in the management of patients with high-grade EOC.

16.Line 316: Please delete "in"

17.Line 319 and line 320: Please re-write: That means, the applied assay does not 319 require to generate a priori information about specific tumor-mutations and evaluates

18.Line 321-322: Please re-write: We can confirm the reported high sensitivities to distinguish between high-grade EOC and healthy controls.

19.Line 330-line 334: Please clarify by re-writing

20.Line 337-line 339: Please re-write to clarify.

21.Line 343-345: Sentence needs to be re-written.

22.Line 346: Please change “proved evidence”

23.Line 352: please change “to” into “in”

24.Line 363: please change “effectivity ” into “effectiveness”

25.Line 366: please change “be valid ” into “a valid”

26.Line 367-369: Please rewrite

27.Line 371: early-stage primary tumors, and longitudinal blood samples of each patient, as well as the small sam

Please add the highlight.

Author Response

Reviewer 1:

I am impressed by the work in this manuscript and am recommending it for publication after revising modifications. à thanks!

I have indicated where certain textual are needed as follows.

The authors need to provide the power analysis for this study. -->  We included the following sentence to ll.92-95: ” A power analysis according to the primary goal was performed with the following assumptions: mean difference to normal cohort: two-fold with a sigma of 4 and an alpha of 0.01, which resulted in a minimum of 66 patients to reach a power of 0.95..”

Please explain why the control results in the left and middle panels of Figure 1 seem different. à the scale of the y-axis is different between both graphs.

I recommend that they enlist the assistance of a skilled English-speaking editorial writer. --> conducted, and some changes were advised and applied.

I would like to see a little more discussion of how this approach could be used in monitoring treatment, including costs in time and money. à we added the following sentence to ll. 385-390: “This approach would demand timely results with cost effectiveness. Both is applicable with the CNI-Score, as the turnaround times between blood-draw and result reporting is about five days, with a total of three days for the entire laboratory analysis. Costs for CNI-Score determination is roughly about 1/10th-1/5th of the cost of one cycle of therapy with e.g. a check-point inhibitor, which would also be considered as representable. ”

Detail revision:

line 33: pact of the CNI-Score to detect epithelial ovarian cancer (EOC) and monitor treatment, respectively. please change "and" into "so that it could be used to" à thanks, for the suggestion. changed

2.Line 79-81: Please rewrite; sentences were re-written: “Furthermore, copy-number differences can be detected by low-coverage whole genome sequencing, which is cost-effective and is less prone to false positive findings and quantification errors due to sequencing errors. Moreover, it requires less extensive bioinformatic processing of the raw data (7, 8).”

3.line 116-119: Please re-write this sentence so that it makes sense.--> thank you very much for this annotation, sentence was changed to:” As The cell-free tumor DNA (ctDNA) assay, is an accredited (ISO/EN 14189:2018) clinical grade assay, which is based on the quantification of ctDNA by using copy-number variances (a hallmark of malignant cells), detected in plasma of patients by shallow se-quencing (9).”

4.Line 91: primary advanced  Please rewrite, 5.Line 92: num-eligible, or non-platinum eligible relapsed high-grade EOC  Please rewriteà thank you very much for this annotation, sentence was changed to:” The following inclusion criteria had to be fulfilled: patients with primary advanced (FIGO IIIB-IVB) high-grade, or platinum-eligible, or non-platinum eligible relapsed high-grade EOC and a signed informed consent form.”

6.Line 99: Please check word “were”à checked, correct!

7.Line 101: Please check word “were” à checked, correct!

8.Line 110: Please check word “were” à checked, correct!

9.Line 111: Please check word “does”à do not find word “does” in line 111?!

10.Line 116-118: Please re-write this sentence so that it makes sense.--> Lines do not include a full sentence?!

11.Page 5: Please enter percentage numbersà the percentage sign “%” is given in brackets at the top of each column, and the values in the brackets represent the percentage value in each cell.

12.Line 240: Please check: one patient that was collected 1814 days (5 years) after CTXà correct.

13.Line 244: Please check: progression was not clinically diagnosedà correct!

14.Line 279: Please delete "rather"à added

15.Line 304-305: Please re-write this sentence and indicate how.: On the other hand, presented data give rise for potential application of the 304 CNI-Score in the management of patients with high-grade EOC.--> follows later on up to line 356 ff.

16.Line 316: Please delete "in"à removed

17.Line 319 and line 320: Please re-write: That means, the applied assay does not 319 require to generate a priori information about specific tumor-mutations and evaluates à thanks, sentence changed to:” That means, the applied assay does not require a priori information about specific tumor-mutations and evaluates the whole-genome instead of only prespecified target sites [11].”

18.Line 321-322: Please re-write: We can confirm the reported high sensitivities to distinguish between high-grade EOC and healthy controls. à Sentence changed to:” With the data presented, Wwe can confirm the reported high sensitivities to distinguish between high-grade EOC and healthy controls.”

19.Line 330-line 334: Please clarify by re-writingà Thanks, sentence was rewritten to:” Furthermore, SNP-targeting high-throughput sequencing approaches that are deemed ul-trasensitive are often more time-consuming, costly, and computationally intensive and were able to detect tumor-specific SNPs in the circulation of 78% of ovarian cancer patients with tumor stages III and IV [24].”

20.Line 337-line 339: Please re-write to clarify.--> thanks, sentence was changed to:” Due to the high sensitivity and specificity of the CNI-Score to we were able to distinguish EOC from healthy cases. Moreover, follow-up and treatment might be altered in patients with high-grade EOC, due to CNI-Score analyses.”

21.Line 343-345: Sentence needs to be re-written.--> sentence changed to:” The “Rustin trial” changed standard follow-up after the end of 1st -line chemotherapy eliminating regularly CA125 determination, as early detection and early onset of systemic treatment of recurrence did not improved survival [29].”

22.Line 346: Please change “proved evidence”-_> changed to “showed”

23.Line 352: please change “to” into “in”à changed

24.Line 363: please change “effectivity ” into “effectiveness”à changed

25.Line 366: please change “be valid ” into “a valid”à changed

26.Line 367-369: Please rewriteà changed

27.Line 371: early-stage primary tumors, and longitudinal blood samples of each patient, as well as the small sam à do not understand the remark.

Please add the highlight. --> Added to ll. 386-389:” We could show that CNI-Score levels are neither significantly different between patients in varying treatment situations, nor between different biologic characteristics, thus the CNI-Score might be a useful marker in all patients with EOC.”

Reviewer 2 Report

General comment:

  1. The author using sequencing technology to screen ctDNA of high-grade EOC patient blood sample of various conditions. They found a non-significant value of diagnostic power in study groups (upfront vs. CTX vs. follow-ups). Although this data discourages the value of using ctDNA liquid biopsy in EOC diagnosis, reviewer consider this as important data to the field.
  2. In the discussion, the authors mentioned: In addition, ctDNA seems to have better diagnostic accuracy and especially higher specificity than CA-125 and/or RMI (risk of malignancy) scoring-albeit not proven in the present paper-, and others propose it as presurgical biomarker to differentiate benign adnexal masses from malignant disease.
  3. --->But only 23 patients were in the upfront group, what's their CNI-scores compare with healthy group, this part should be mentioned.
  4. The discussion result read awkward to reviewer. For example, the first paragraph of discussion, author stated “(page 10)….presented data give rise for potential application of the CNI-Score in the management of patients with high-grade EOC.” Since the data presented showed insignificant difference between groups, why it still valuable?
  5. CNI-Scores above this threshold were detected in 18/23 primary tumors (78%), 35/42 of platinum-eligible recurrent (83.3%) and 19/22 of non- platinum-eligible recurrent (82.6%) samples, respectively. --->How it correlated to clinical subgroup should be mentioned inside the discussion. 
  6. Does the CNI-scores change during the treatment course, like other tumor markers.
  7. The CNI-Score may need to be correlated with the clinical tumor markers to show the clinical application of CNI-Score. 
  8. To reviewer point of view, if there’s no superior of new diagnosis method than conventional method, then discard new one. The author need clarify this point.

Author Response

The author using sequencing technology to screen ctDNA of high-grade EOC patient blood sample of various conditions. They found a non-significant value of diagnostic power in study groups (upfront vs. CTX vs. follow-ups). Although this data discourages the value of using ctDNA liquid biopsy in EOC diagnosis, reviewer consider this as important data to the field.--> thank you!

In the discussion, the authors mentioned: In addition, ctDNA seems to have better diagnostic accuracy and especially higher specificity than CA-125 and/or RMI (risk of malignancy) scoring-albeit not proven in the present paper-, and others propose it as presurgical biomarker to differentiate benign adnexal masses from malignant disease. --->But only 23 patients were in the upfront group, what's their CNI-scores compare with healthy group, this part should be mentioned.--> Thank you for this suggestion. We changed the sentence: “With the integration of just 23 patients with primary diagnosis of high grade EOC, we were able to differentiate between those patients and healthy controls with a sensitivity of 78% at a specificity of 99.6%. When adding patients with recurrent disease to the patients with primary diagnosis of high grade EOC the differentiation between EOC and healthy controls the sensitivity was 83%.”

The discussion result read awkward to reviewer. For example, the first paragraph of discussion, author stated “(page 10)….presented data give rise for potential application of the CNI-Score in the management of patients with high-grade EOC.” Since the data presented showed insignificant difference between groups, why it still valuable? --> sentence added:” Due to the fact, that CNI-Score levels are neither significantly different between patients in varying treatment situations (primary diagnosis/ platinum-eligible/non-platinum eligible), nor between different biologic characteristics (BRCA status) the CNI-Score might be a useful marker in all patients with EOC.”

CNI-Scores above this threshold were detected in 18/23 primary tumors (78%), 35/42 of platinum-eligible recurrent (83.3%) and 19/22 of non- platinum-eligible recurrent (82.6%) samples, respectively. --->How it correlated to clinical subgroup should be mentioned inside the discussion. --> please find the sentence above, which includes the demanded remark.  

Does the CNI-scores change during the treatment course, like other tumor markers.--> we do not know, and had included it already to the limitations section, that longitudinal samples were not taken in this study.

The CNI-Score may need to be correlated with the clinical tumor markers to show the clinical application of CNI-Score. --> added to the sentence l. 387-388: “…”(e.g. comparison to other tumor markers like CA125)”…

To reviewer point of view, if there’s no superior of new diagnosis method than conventional method, then discard new one. The author need clarify this point. --> sentence added:” New markers for diagnostics and therapy monitoring that can show incremental value to current standard options might be useful tools in patient management if those outperform current standard options. ”

Reviewer 3 Report

Dear authors:
Congratulations on this new and well-developed work. From a clinical point of view, the results are very promising, especially for professionals involved in the treatment of ovarian cancer.
Excuse my ignorance in basic research but I have a series of questions that I would like you to answer to better assess your work.
1.- I believe that a better explanation of the CNI score calculation methodology is necessary so that a clinician can understand it and not only refer to previous publications.
2.- Do you think these results could be reproduced in other types of non-serous ovarian tumors?
Regarding the rest of the methodology and presentation of results, nothing to object, it seems perfect to me.
Regarding the discussion, although it also seems entirely correct to me, with clear references to surgery and control of relapses that seem very innovative to me. I cannot assess it in its entirety because the references from number 19 onwards are missing, therefore I cannot recommend the publication with the current manuscript format.

Author Response

Congratulations on this new and well-developed work. From a clinical point of view, the results are very promising, especially for professionals involved in the treatment of ovarian cancer.

Excuse my ignorance in basic research but I have a series of questions that I would like you to answer to better assess your work.

1.- I believe that a better explanation of the CNI score calculation methodology is necessary so that a clinician can understand it and not only refer to previous publications. --> We double-checked the CNI Score calculation methodology and found no further information to provide to better understand the procedure conducted.  

2.- Do you think these results could be reproduced in other types of non-serous ovarian tumors?--> we do not now at all, but we have included thi point to the outlook for future study activities ll. 387-390:” Current data give rise to  future studies that aim to prove the versatility of the CNI-Score in other entities (e.g. non-high grade EOC) and in comparison to established markers (e.g. to other tumor markers like CA125) and to  for therapy and recurrence monitoring in EOC.”

Regarding the rest of the methodology and presentation of results, nothing to object, it seems perfect to me.--> thanks!

Regarding the discussion, although it also seems entirely correct to me, with clear references to surgery and control of relapses that seem very innovative to me. I cannot assess it in its entirety because the references from number 19 onwards are missing, therefore I cannot recommend the publication with the current manuscript format. --> sorry. We did add the missing references.